# OBJECT HIERARCHY GUIDES VISUAL RELATIONSHIP REASONING

## ABSTRACT

Visual relationship, commonly defined as tuple consisting of subject, predicate and object, plays an important role in visual scene understanding, Most existing works are dedicated to generating discriminative predicate representation for the detected objects, based on their appearances, relative positions and global context. However, these global representations are inherently ambiguous and confounded, often capturing irrelevant contextual information. To address this problem, we propose to leverage object hierarchy to infer visual relationships. Our core insight is that a seemingly holistic object-level interaction can be resolved into a set of precise part-level interactions via an object hierarchy. Compared with object-level interaction, part-level interaction not only has lower visual variability, but also provides accurate guidance for model understanding predicates. To this end, we introduce Hierarchical Inference Network (HINet). Specifically, we first construct more robust and discriminative predicate representations by dynamically fusing global object-level and local part-level representations. We then design a structured constraint on predicate representations by explicitly constructing correlations between object-level and part-level interactions, thereby guiding the model to focus on the key information of the current interaction. Through the collaborative processing of these strategies, our HINet transcends the superficial learning of visual relations from objects and predicates, adopting a structured reasoning approach to explore their essence. Experiments have demonstrated the effectiveness of our method. Furthermore, it exhibits strong versatility and can be efficiently integrated with various existing models to enhance their performance.

## 1 INTRODUCTION

Visual relationship plays an important role in visual scene understanding, which is commonly defined as a tuple consisting of subject, predicate and object (Lu et al., 2016). Such an interactive tuple has potential applications in many vision tasks such as visual question answering (Qian et al., 2024; Lin et al., 2024; Gao et al., 2024), visual relationship detection (Li et al., 2024c; Lu et al., 2016; Liang et al., 2018) and scene graph generation (Li et al., 2024b; Zhao et al., 2024; Lin et al., 2024). In recent years, object-centric techniques have matured significantly, thanks to learned visual representations and advances in object detection. However, the modeling and understanding of predicates remain a critical bottleneck, hindering progress in downstream tasks.

Most existing works follow a two-stage paradigm that detects objects first and then predicts their predicates (Krishna et al., 2017; Tang et al., 2019). They are dedicated to generating discriminative predicate representations for the detected objects, based on their appearances, relative positions, and global context (Xu et al., 2017; Zellers et al., 2018; Li et al., 2021). However, these global representations are inherently ambiguous and confounded, often capturing irrelevant contextual information. For example, the union region of "man" and "bat" may simultaneously include the information for "man-hold-bat", "man-wear-shirt" and "man-has-hand", making it challenging to identify the key interaction. Meanwhile, these global representations are often difficult to capture the subtle differences between some similar predicates. For example, "carrying" (a person having an object in their hands) and "holding" (a person supporting an object in their hands) (Li et al., 2023). These problems motivate the key question: *how to extract key information from ambiguous and confounded global representations to facilitate accurate predicate inference by the model*?

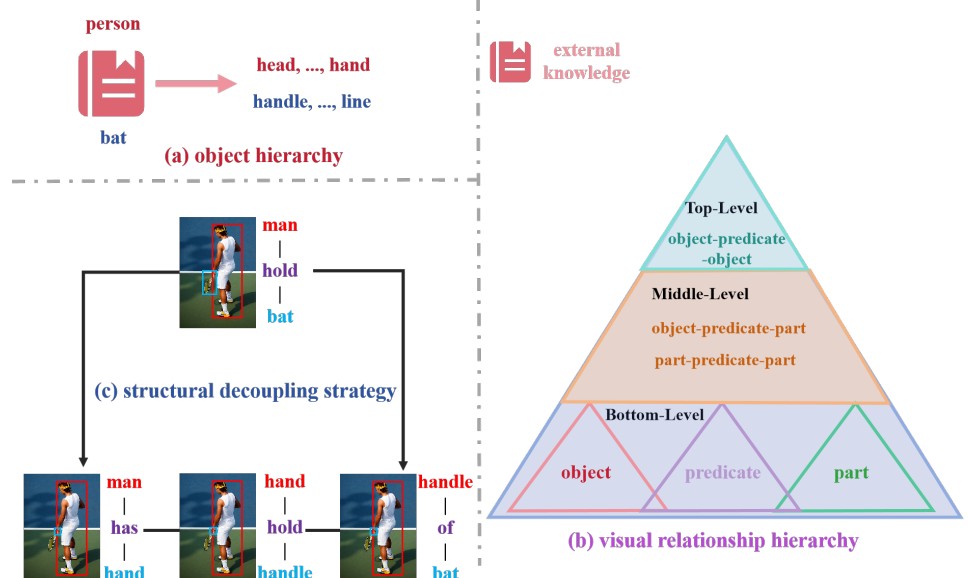

Figure 1: The overview of our idea. **(a)** We use external knowledge and object category to generate part categories; **(b)** A visual relationship hierarchy caused by object hierarchy. **(c)** The structural decoupling strategy of "man-hold-bat" caused by visual relationship hierarchy.

To address this problem, we propose to utilize object hierarchy (or the combination of object and its parts) to accurately infer predicates (as shown in Fig.1.**a**). **Our core insight** is that a seemingly holistic object-level interaction can be resolved into a set of precise part-level interactions via object hierarchy. The effectiveness of our idea stems from the two key advantages. First, compared with object-level interaction, part-level interaction can not only capture fine-grained information but also exhibit less visual variability, thereby providing more robust and discriminative predicate representations. Second, object hierarchy can provide a structured decoupling of visual relationships, allowing the model to focus on the key information of interaction. This idea stems from an interesting phenomenon: although all visual relationships are composed of objects and predicates, some of them can be effectively represented by the combination of others. For example in Fig.1.**c**, "man-hold-bat" can be effectively represented by the combination of "man-has-hand", "hand-hold-handle" and "handle-of-bat". From the hierarchical knowledge perspective, we summarize this phenomenon as a visual relationship hierarchy caused by object hierarchy (as shown in Fig.1.**b**). In this structure, any higher level entity can be represented by a combination of lower level entities.

Compared with the existing methods that simply represent visual relationships through objects and predicates (Xu et al., 2017; Dai et al., 2017; Li et al., 2017), we reveal a reasoning association between different visual relationships via object hierarchy. Meanwhile, unlike any existing works on hierarchy in SGG, our method neither performs explicit hierarchical modeling of predicates (Tang et al., 2019; Zhou et al., 2020; Zhang et al., 2024) nor implicitly optimizes the context information using object parts (Lu et al., 2018; Tian et al., 2020; Dong et al., 2021). It aims to guide the model to perform a structured decoupling of relationships through the structural prior of object hierarchy, thereby enabling the model to better understand visual relationships. For object-level interaction, this strategy can effectively guide the model to focus on the key information; for part-level interaction, this strategy provides a novel structured constraint for optimizing predicate representation.

To realize this idea, we introduce **H**ierarchical **I**nference **Net**work (**HINet**), which utilizes object hierarchy to guide the model to accurately infer predicates through two collaborative strategies. First, to enhance the ability of predicate representation to perceive details, we propose **O**bject-**P**art **H**ybrid **P**erception strategy (**OPHP**). It dynamically integrates global object-level representations with local part-level representations to construct more robust and discriminative predicate representations. This hybrid representation alleviates the ambiguity of predicate representation by capturing global context and local details. Second, to alleviate the problem of confounded information in global rep-

resentations, we propose **H**ierarchical **C**onsistency **R**easoning strategy (**HCR**). It introduces a novel structured constraint for predicate representation by explicitly constructing the correlations between entities at different levels in our visual relationship hierarchy (as shown in Fig.1.**b**). By aligning the representations of entities at different levels in the embedding space, this structured prior can effectively guide the model to focus on the key information of the current interaction.

To verify the effectiveness of our method, we utilize Visual Genome (Krishna et al., 2017), Open Images V6 (Kuznetsova et al., 2020), V-COCO (Gupta & Malik, 2015) and HICO-DET (Chao et al., 2018). Extensive experimental results demonstrate the effectiveness of our ideas. **Our main contributions** are summarized as follows: **1)** We notice the ambiguity and information confounding of the predicate representation used in the existing methods, and propose a novel structural decoupling strategy to solve them by utilizing object hierarchy. **2)** We introduce a novel Hierarchical Inference Network (HINet). It dynamically fuses multi-level representations via OPHP strategy to enhance predicate representation discriminability, and imposes a novel structured constraint on the model through HCR strategy, guiding it to focus on key interaction information. **3)** Our method achieves competitive or state-of-the-art performance on various scene graph benchmarks. More importantly, our idea is model-agnostic and can be applied to several existing SGG models.

## 2 OUR METHOD

Our proposed HINet is designed to address the challenges of ambiguity and information confounding in predicate representation. An overview of our framework is illustrated in Fig.2. Following the standard two-stage paradigm (Zellers et al., 2018; Li et al., 2021), we first generate a set of object and relationship proposals. Then, we introduce a object hierarchy generation step to build a part-level knowledge. This hierarchy then enables two core components of our model: **OPHP** strategy for robust feature extraction, and **HCR** strategy for structured decoupling visual relationships.

### 2.1 PROPOSAL AND OBJECT HIERARCHY GENERATION

**Proposal Generation.** Follow the settings of existing works (Zellers et al., 2018; Li et al., 2021), we first utilize an object detector network (e.g., Faster R-CNN (Ren et al., 2015)) to generate a set of object and relationship proposals. The object proposals are taken directly from the detection output with their categories and classification scores, while the relationship proposals are generated by forming ordered pairs of all the object proposals.

**Object Representation Calculation.** For object representation, the calculating method is consistent with BGNN (Li et al., 2021). Specifically, for the $i$-th object proposal, we denote its convolution feature as $v_i$, its bounding box as $b_i$ and its detected class as $c_i$. Then, we can utilize $b_i$ to calculate geometric feature $g_i$, and utilize $c_i$ to calculate semantic feature $w_i$. Finally, the object representation $o_i$ is computed as

$$o_i = f_o(v_i \oplus g_i \oplus w_i), \qquad (1)$$

where $f_o$ is a fully-connected network, and $\oplus$ is the concatenation operation.

**Object Hierarchy Generation.** Following the settings in Motif (Zellers et al., 2018), we classify all detected object categories into two super-types: part-level object (such as: "foot", "hand", "leaf", "branch") and non-part object (such as: "human", "tree"). For all non-part objects, we utilize LLM (such as: GPT-4) to collect their part categories. The relevant prompts can be found in Sec.B. For the $i$-th object proposal, this process transforms a single class label $c_i$ into a set $h_i = \{h_i^k\}_{k=0}^{k_i}$, where $h_i^0 = c_i$, $h_i^z$ represents the $z$-th part category of $c_i$ and $k_i$ represents category $c_i$ has $k_i$ parts. It is worth noting that if $c_i$ belongs to part-level object, $h_i = \{h_i^0\}$. Meanwhile, to ensure fair comparison and stability of the model, we additionally utilize **WordNet** to collect the part categories.

Through this strategy, we generate a set of object and relationship proposals for each image, calculate the object representation for each object proposal, and generate object hierarchy for each object category. Based on it, we propose **OPHP** strategy to calculate the predicate representation.

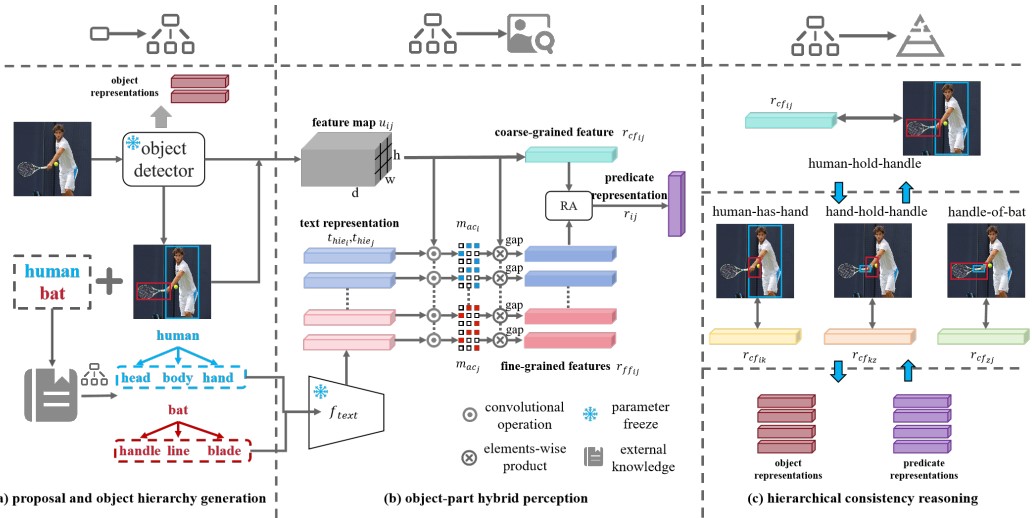

Figure 2: Illustration of overall pipeline of our HINet model. It contains three modules: a) proposal and object hierarchy generation; b) object-part hybrid perception; c) hierarchical consistency reasoning. **RA** denotes representation aggregation strategy; **external knowledge** represents the method of generating object hierarchy; $m_{ac_{ij}}$ denotes the spatial activation map.

## 2.2 OBJECT-PART HYBRID PERCEPTION

To alleviate the ambiguity inherent in coarse global features, our OPHP strategy constructs a richer predicate representation by dynamically fusing global context with fine-grained local details. This process involves three steps: coarse-grained feature calculation (global representation), fine-grained feature calculation (local representation) and representation aggregation (RA).

**Coarse-grained feature calculation.** In the process of human inferring visual relationships, we first take a glance at the visual appearance and holistic position of objects. These coarse-grained features usually determine the overall interpretation of visual relationships (Gao et al., 2021). In this paper, our method for calculating coarse-grained feature is consistent with the previous works for representing predicates (Li et al., 2021; 2022b; 2024a). Specifically, given the relationship proposal from object $i$ to $j$, its coarse-grained feature $r_{cf_{ij}}$ can be computed by

$$r_{cf_{ij}} = f_{cv}(v_i \oplus v_j) + f_{cg}(g_i \oplus g_j), \tag{2}$$

where $f_{cv}$ and $f_{cg}$ are two fully-connected networks that integrate the convolution features and geometric features of objects $i$ and $j$.

**Fine-grained feature calculation.** To make up for the lack of fine-grained information in global representation, we utilize object hierarchy to capture more fine-grained features. A major reason is that the fine-grained features that infer predicates rely on are generally strongly correlated with parts. For example, "stand on" depends on the state of "leg".

To calculate the fine-grained features, which are highly correlated with parts, we first map object hierarchy from category label to text embedding. In this paper, we try two methods to calculate text embedding, which are GloVe word embedding (Pennington et al., 2014) and CLIP text embedding (Radford et al., 2021), respectively. The effectiveness of the former has been proved in a large number of works (Zhang et al., 2021), while the latter is considered by some researchers to contain better semantic knowledge (Yang et al., 2023). For the $i$-th object proposal, the text embedding of its object hierarchy $t_{hie_i} = \{t^k_{hie_i}\}^{k_i}_{k=0}$ is formulated as

$$t^z_{hie_i} = f_{text}(h^z_i), \tag{3}$$

where $f_{text}$ represents the method of mapping the $z$-th category label in object hierarchy to text embedding.

Given the relationship proposal from object $i$ to $j$, we then denote the convolution feature map obtained by their union-box as $u_{ij} \in \mathbb{R}^{d \times h \times w}$. And every text embedding of $t_{hie_i}$ and $t_{hie_j}$ will

be used to perform a convolutional operation on the feature map and can obtain two sets of spatial activation maps $m_{ac_i} = \{m_{ac_i}^k\}_{k=0}^{k_i}$ and $m_{ac_j} = \{m_{ac_j}^k\}_{k=0}^{k_j}$:

$$m_{ac_i}^z = sigmoid(u_{ij}, \odot t_{hie_i}^z) \tag{4}$$

where $m_{ac_i}^z \in R^{h \times w}$ and $\odot$ denotes the convolutional operation. Each value in the spatial activation map represents how likely this local region contains the corresponding component. Since the last operation is a sigmoid function, these values are bounded between [0, 1].

Finally, we utilize these spatial activation maps as region-based attention weights and process them with the previously obtained convolution feature map to calculate fine-grained features. And for the relationship proposal from object $i$ to $j$, we can get two sets of fine-grained features $r_{ff_i} = \{r_{ff_i}^z\}_{z=0}^{z=k_i}$ and $r_{ff_j} = \{r_{ff_j}^z\}_{z=0}^{z=k_j}$:

$$r_{ff_i}^z = gap(u_{ij} \otimes m_{ac_i}^z), \tag{5}$$

where $\otimes$ denotes elements-wise product and $gap$ is the global average pooling function.

**Representation Aggregation (RA).** Finally, we propose a representation aggregation (RA) strategy to fuse the coarse-grained and fine-grained features to represent predicates. Recognizing that not all parts are relevant to a given interaction, for the relationship proposal from object $i$ to $j$, our RA strategy first compute two sets of relevance score $s_{i \to j} = \{s_{i \to j}^k\}_{k=0}^{k_i}$ and $s_{j \to i} = \{s_{j \to i}^k\}_{k=0}^{k_j}$ by

$$s_{i \to j}^z = cos\_sim(v_j, r_{ff_i}^z), \tag{6}$$

where $s_{i \to j}^z$ represents how important $z$-th part of object $i$ is for interacting with object $j$. We then treat them as weights to aggregate fine-grained features, and the predicate representation $r_{ij}$ can be calculated by the following formula:

$$r_{ij} = f_r(r_{cf_{ij}} \oplus \frac{1}{k_i} \sum_{z=0}^{k_i} s_{i \to j}^z \cdot r_{ff_i}^z \oplus \frac{1}{k_j} \sum_{z=0}^{k_j} s_{j \to i}^z \cdot r_{ff_j}^z) \tag{7}$$

where $f_r$ is a fully-connect network that integrates the coarse-grained and fine-grained features.

## 2.3 HIERARCHICAL CONSISTENCY REASONING

To address the information confounding problem, where predicate features are polluted by irrelevant context, we introduce **HCR** strategy. Our idea stems from an interesting phenomenon: although all visual relationships are composed of objects and predicates, some of them can be effectively represented by the combination of others. We summarize this phenomenon as a visual relationship hierarchy caused by object hierarchy (as shown in Fig.1.**b**). In this structure, higher level entities can be represented by a combination of lower level entities. It is essentially a structured decoupling strategy for visual relationships, which can effectively guide the model to focus on key information. More detailed explanations can be found in our appendix. Based on it, we design **HCR** strategy to provide a structured constraint for predicates, including the following two methods:

**Bottom-to-others.** At the center of this method lies the definition of visual relationship: visual relationship is a pair of localized objects via a predicate (Lu et al., 2016). It means that the visual appearance in union-box of the subject and the object is more like the visual relationship representation (or, the representation of triplet), rather than the predicate representation. In other words, the connection of subject representation, predicate representation and object representation should be similar to visual relationship representation. Formally, they follow the following formula:

$$r_{vre} \approx r_{sub} + r_{pre} + r_{obj}, \tag{8}$$

where $r_{vre}, r_{sub}, r_{pre}, r_{obj}$ represent visual relationship representation, subject representation, predicate representation and object representation, respectively.

We then construct it in embedding space. Concretely, for the relationship proposal from object $i$ to $j$, this method takes as input the object representations, predicate representation and coarse-grained feature (treat it as visual relationship representation). And Eq.8 can be represented as

$$r_{cf_{ij}} = o_i + r_{ij} + o_j. \tag{9}$$

It reflects the correlations between bottom-level and other levels in our hierarchy, thus it is applicable to all relationship proposals. Based on it, we can provide a novel constraint $L_{bto}$ for optimizing predicate representation:

$$L_{bto} = max(cos\_sim(r_{ij}, r_{cf_{ij}} - o_i - o_j)), \qquad (10)$$

where $cos\_sim$ represents the cosine similarity between representations and Eq.10 represents maximize the cosine similarity at the representations of Eq.9.

**Middle-to-top.** This method stems from that some visual relations can be effectively represented by the combination of others. The associations between objects and parts make us can always find a set of object-predicate-part, part-predicate-part to represent object-predicate-object. For example, "person-stand on-road" can be inferred by: "person-has-foot", "foot-stand on-surface" and "surface-of-road". Here, we assume that there is a set of visual relationships satisfying the above description, where $S$ represents object-predicate-object, $K$ represents part-predicate-part, and $P_1$, $P_2$ represent object-predicate-part. And then they follow the following formula:

$$S \approx P_1 + K + P_2. \qquad (11)$$

We then construct it in embedding space. Concretely, given the relationship proposal from object $i$ to $j$, if they are non-part objects, we will construct this correlation by the following steps.

**Step 1: part selection.** Among all predicates, some predicates are strongly related to parts. For example, "stand on" is strongly correlated related to "foot" and "surface". For these predicates, we manually filter their parts to construct the correlation in Eq.11. And for other predicates, we randomly select the parts of their subject and object. It is worth mentioning that, for each relationship proposal, only one part is selected for each object.

**Step 2: part proposal generation.** We then generate the proposals of the selected parts. Due to the lack of relevant annotations, we utilize the idea of RegionCLIP (Zhong et al., 2022) to complete this task. For the relationship proposal from object $i$ to $j$, we denote the proposals of their selected parts be $k$, $z$, respectively. Based on it, we can get new three sets of relationship proposals: $i$ to $k$, $k$ to $z$ and $z$ to $j$.

**Step 3: hierarchical representation generation.** Finally, we can calculate their representations $r_{cf_{ik}}, r_{cf_{kz}}, r_{cf_{zj}}$ by Eq.2. And Eq.11 can be represented as

$$r_{cf_{ij}} = r_{cf_{ik}} + r_{cf_{kz}} + r_{cf_{zj}}. \qquad (12)$$

It reflects the correlations between the middle-level and top-level in our hierarchy. It makes the model's understanding of the visual relationship no longer a simple combination of objects and predicates, and provides a detailed reasoning way for the model to understand the visual relationship. Based on it, we can provide a novel constraint $L_{mtt}$ for optimizing the representations in Eq.12:

$$L_{mtt} = max(cos\_sim(r_{cf_{ij}}, r_{cf_{ik}} + r_{cf_{kz}} + r_{cf_{zj}})). \qquad (13)$$

In general, this strategy designs a novel structured constraint for predicate representation by explicitly constructing the correlations between entities at the different levels in our visual relationship hierarchy. Formally, our HCR strategy is similar to TransE (Bordes et al., 2013) in knowledge graph. By aligning the representations of entities at different levels in the embedding space, this structured prior can effectively guide the model to focus on the key information of the current interaction. More details of this strategy can be found in Sec.C, including design idea (in Sec.C.1), overall structure (in Sec.C.2) and visualized results (in Sec.C.2).

## 2.4 LEARNING STRATEGY

**Prediction.** To predict the object and predicate, we introduce two linear classifiers. For predicate, our classifier integrates the predicate representation $r_{ij}$ and a class frequency $q_{ij}$ prior for classification (Zellers et al., 2018). The distribution of predicate $pred_{r_{ij}}$ is computed as

$$pred_{r_{ij}} = softmax(W_{rel}r_{ij} + q_{ij}), \qquad (14)$$

where $W_{rel}$ is the parameter of predicate classifier. For object, our classifier takes as input the object representation $o_i$. The distribution of object $pred_{o_i}$ is computed as

$$pred_{o_i} = softmax(W_{obj}o_i), \qquad (15)$$

where $W_{obj}$ is the parameter of object classifier.

**Training Loss.** To train our HINet model, we design a multi-tasks loss $L_{total}$ of three components, including $L_p$ for predicate classification, $L_o$ for object classification, $L_{hcr}$ for HCR strategy. Formally,

$$L_{total} = L_p + \lambda_o L_o + \lambda_{hcr} L_{hcr} \tag{16}$$

where $\lambda_o$, $\lambda_{hcr}$ are weight parameters for calibrating the supervision from each sub-task. Here $L_p$, $L_o$ are the standard cross entropy loss for multi-class classification (foreground categories plus background). The loss of HCR ($L_{hcr}$) is composed by two components: $L_{hcr} = L_{bto} + L_{mtt}$.

## 3 EXPERIMENTS

In general, to fully verify the effectiveness of our idea, we conduct experiments on four datasets. Due to the page limitations, in this section, we only show the experimental results of Visual Genome, and more detailed results can be found in our appendix (Open Images V6 in Sec.D, HICO-DET in Sec.D.2 and V-COCO in Sec.D.2). The visualized results also can be found in Sec.D, and the related works can be found in Sec.A. The LLMs usage description in our work can be found in Sec.E.

### 3.1 EXPERIMENTS CONFIGURATION OF VISUAL GENOME

**Dataset Details.** We utilize Visual Genome (VG) dataset to verify the effective of our idea. It consists of 108,073 images, including tens of thousands of unique object and predicate categories. In our experiments, we follow the most commonly used data splits proposed by (Xu et al., 2017; Zellers et al., 2018). The 150 most frequent object categories and the 50 most frequent predicate types are adopted for evaluation.

**Evaluation Protocol.** Following the most existing works, we evaluate our model on three sub-tasks: 1) predicate classification (PredCls); 2) scene graph classification (SGCls); 3) scene graph generation (SGGen). In each task, following existing works (Zellers et al., 2018; Li et al., 2021; Hayder & He, 2024), we take **recall** (**R@K**), **mean recall** (**mR@K**) and **overall mean** (**M@K**) as evaluation metrics.

**Implementation Details.** In general, we introduce our implementation details from the following aspects: **1) object detector**: in our experiment, following the previous works (Li et al., 2021; 2024a), we adopt the pre-trained Faster-RCNN with ResNeXt-101-RPN (Xie et al., 2017) as object detector to obtain the object and relationships proposals; **2) parameter setting**: our experiments were performed on three 3090 GPUs. The batch size and initial learning rate are set to 9 and 0.024, respectively. And $\lambda_o$, $\lambda_{hcr}$ in Eq.16 set to 1,5, respectively. Our model is optimized by the Adam algorithm with the momentum of 0.9 and 0.999. **3) use of large models**: in our method, we utilize the "gpt-4-0613" API to generate part categories, and the relevant prompt can be found in our appendix. And CLIP (VIT-B/16) (Radford et al., 2021) is used to provide text embedding.

### 3.2 COMPARISONS WITH STATE-OF-THE-ART METHODS

According to the research content of the two-stage scene graph generation methods in recent years, the SOTA methods we compared are mainly divided into two aspects:

**1) The effectiveness of our method.** Some works try to improve the predicate representation by utilizing the context information of the scene, which have the similar purpose to us. It is worth mentioning that our HINet is based on BGNN (shadow background in Tab.1). We directly compare our method with them, and Tab.1 shows the comparison results of these methods. From Tab.1, we have the following observations: **1)** our proposed method has achieved significant improvement on mR@K. More specifically, our method outperforms the EdgeSGG by **4.1%**, **3.6%** and **3.6%** at mR@100 on PredCls, SGCls and SGGen, respectively. Because the VG dataset has an imbalanced data distribution, mR@K, which prefers tail predicates, can be said to be more reliable than R@K metrics that focus on common predictions with abundant samples (Li et al., 2021; Zhang et al., 2020). **2)** Although our R@K decreases slightly, M@K has superior performance. It indicates that our method doesn't increase mR@K at the expense of reducing R@K, which is different from some methods (e.g., DRM). These results prove that our method is effective for most predicates.

| Models | PredCls | | | SGCls | | | SGGen | | |
|---|---|---|---|---|---|---|---|---|---|
| | mR@50/100 | R@50/100 | M@50/100 | mR@50/100 | R@50/100 | M@50/100 | mR@50/100 | R@50/100 | M@50/100 |
| VCTree (Tang et al., 2019) | 17.9/19.4 | 66.4/68.1 | 42.1/43.7 | 10.1/10.8 | 38.1/38.8 | 24.1/24.8 | 5.9/8.0 | 27.9/31.3 | 16.9/19.6 |
| Unbiased (Tang et al., 2020) | 25.4/28.7 | 47.2/51.6 | 36.3/40.1 | 12.2/14.0 | 25.4/27.9 | 18.7/20.9 | 9.3/11.1 | 19.4/23.2 | 14.3/17.1 |
| MSDN (Li et al., 2017) | 19.2/20.5 | 65.0/66.7 | 42.1/43.6 | 11.6/12.6 | 38.9/39.8 | 25.2/26.2 | 7.7/9.0 | 30.3/33.3 | 19.0/21.1 |
| GPS-Net (Lin et al., 2020) | 15.2/16.6 | 65.2/67.1 | 40.2/41.8 | 8.5/9.1 | 37.8/39.2 | 23.1/24.1 | 6.7/8.6 | **31.1/35.9** | 18.9/22.2 |
| SMN (Zellers et al., 2018) | 13.3/14.8 | 65.2/67.1 | 39.2/40.9 | 7.1/7.6 | 35.8/36.5 | 21.4/22.0 | 5.3/6.1 | 27.2/30.3 | 16.2/18.2 |
| BGNN (Li et al., 2021) | 30.4/32.9 | 59.2/61.3 | 44.8/47.1 | 14.3/16.5 | 37.4/38.5 | 25.8/27.5 | 10.7/12.6 | 31.0/35.8 | 20.8/24.2 |
| PPDL (Li et al., 2022b) | 32.2/33.3 | 47.2/47.6 | 39.7/40.4 | 17.5/18.2 | 28.4/29.3 | 22.9/23.7 | 11.4/13.5 | 21.2/23.9 | 16.3/18.7 |
| Nice-Motif (Li et al., 2022a) | 29.9/32.3 | 55.1/57.2 | 42.5/44.7 | 16.6/17.9 | 33.1/34.0 | 24.8/25.9 | 12.2/14.4 | 27.8/31.8 | 20.0/23.1 |
| HetSGG (Yoon et al., 2023) | 31.6/33.5 | 57.8/58.9 | 44.7/46.2 | 17.2/18.7 | 37.6/38.5 | 27.4/28.6 | 12.2/14.4 | 30.0/34.6 | 21.1/24.5 |
| EdgeSGG (Kim et al., 2023) | 34.7/36.9 | 60.1/61.8 | 47.4/49.3 | 17.8/18.8 | 39.1/40.1 | 28.4/29.4 | 13.6/15.8 | 29.7/34.0 | 21.6/**24.9** |
| HIERCOM (Jiang et al., 2023) | 23.9/26.7 | **75.6/79.2** | **49.7/52.9** | 37.5/39.2 | 11.7/12.9 | 24.6/26.0 | 8.2/10.0 | 29.8/32.7 | 19.0/21.3 |
| ST-SGG (Kim et al., 2024) | 28.1/31.5 | 53.9/57.7 | 41.0/44.6 | 16.9/18.0 | 33.4/34.9 | 25.1/26.4 | 11.6/14.2 | 26.7/30.7 | 19.1/22.4 |
| HINet | **38.9/41.0** | 57.6/60.2 | 48.2/50.6 | **21.6/22.4** | **40.8/41.6** | **31.2/32.0** | **17.8/19.4** | 25.6/30.1 | **21.7**/24.7 |

Table 1: The performance of state-of-the-art SGG models on three SGG tasks with graph constraints setting on mR@50/100, R@50/100 and M@50/100 on the VG dataset. The **best** methods are marked according to formats.

| Models | PredCls | | | SGCls | | | SGGen | | |
|---|---|---|---|---|---|---|---|---|---|
| | mR@50/100 | R@50/100 | M@50/100 | mR@50/100 | R@50/100 | M@50/100 | mR@50/100 | R@50/100 | M@50/100 |
| BGNN (Li et al., 2021) | 30.4/32.9 | 59.2/61.3 | 44.8/47.1 | 14.3/16.5 | 37.4/38.5 | 25.8/27.5 | 10.7/12.6 | 31.0/35.8 | 20.8/24.2 |
| HINet | 38.9/41.0 (**+8.1**) | 57.6/60.2 (**-1.1**) | 48.2/50.6 (**+3.5**) | 21.6/22.4 (**+5.9**) | 40.8/41.6 (**+3.1**) | 31.2/32.0 (**+4.5**) | 17.8/19.4 (**+6.8**) | 25.6/30.1 (**-5.7**) | 21.7/24.7 (**+0.5**) |
| PENET (Zheng et al., 2023) | 31.5/33.8 | 68.2/70.1 | 49.8/51.9 | 17.8/18.9 | 39.4/40.7 | 28.6/29.8 | 12.4/14.5 | 30.7/35.2 | 21.5/24.8 |
| HINet + PENET | 39.2/41.1 (**+7.4**) | 65.9/68.4 (**-1.7**) | 52.3/54.8 (**+2.9**) | 23.6/24.2 (**+5.3**) | 41.2/42.6 (**+1.9**) | 32.4/33.4 (**+3.6**) | 18.2/19.9 (**+5.4**) | 26.2/31.1 (**-4.1**) | 22.2/25.5 (**+0.7**) |
| DRM (Li et al., 2024a) | 47.1/49.6 | 43.9/45.8 | 45.5/47.7 | 27.8/29.2 | 27.5/28.4 | 27.6/28.8 | 20.4/24.1 | 19.0/22.9 | 19.7/23.5 |
| HINet + DRM | 49.2/51.9 (**+2.3**) | 44.1/46.4 (**+0.6**) | 46.1/49.2 (**+1.5**) | 29.1/30.5 (**+1.3**) | 30.2/31.4 (**+3.0**) | 29.7/30.9 (**+2.1**) | 22.1/25.9 (**+1.8**) | 21.9/24.1 (**+1.2**) | 22.0/25.0 (**+1.5**) |
| RepSGG (Liu & Bhanu, 2024) | 39.7/43.7 | 27.8/28.8 | 33.7/36.2 | 22.3/27.7 | 17.9/20.3 | 20.1/24.0 | 15.3/18.9 | 12.1/14.6 | 13.7/16.7 |
| HINet + RepSGG | 42.1/47.4 (**+3.7**) | 29.9/31.2 (**+2.4**) | 36.0/39.3 (**+3.1**) | 26.4/31.3 (**+3.6**) | 20.1/22.6 (**+2.3**) | 23.3/27.0 (**+3.0**) | 18.4/21.7 (**+2.8**) | 15.3/16.9 (**+2.3**) | 16.9/19.3 (**+2.6**) |
| RA-SGG (Yoon et al., 2024) | 36.2/39.1 | 62.2/64.1 | 49.2/51.6 | 20.9/22.5 | 38.2/39.1 | 29.5/30.8 | 14.1/17.1 | 26.0/30.3 | 20.2/23.7 |
| HINet + RA-SGG | 42.4/45.3 (**+6.2**) | 61.7/63.9 (**-0.2**) | 52.1/54.6 (**+3.0**) | 23.1/24.3 (**+1.8**) | 38.5/39.4 (**+0.3**) | 30.1/31.9 (**+1.1**) | 18.9/21.3 (**+4.2**) | 25.8/28.2 (**-2.1**) | 22.4/24.8 (**+1.1**) |

Table 2: The performance of our method is combined with other advanced methods on three SGG tasks with graph constraints setting on the VG dataset. The **increased** and **decreased** values are marked according to formats.

**2) Generalization of our method.** Other works design some optimization strategies to help the model learn predicate. These methods also apply to our method. Specifically, we replace the predicate representation in their method with our predicate representation calculated by Eq.7, and add $L_{hcr}$ in Eq.16 to their loss function. Tab.2 shows the comparison results of these methods. From these results, we have the following observations: **1)** Our method is based on BGNN (Li et al., 2021), so we first compare our method with it. It can be seen that our method significantly improves mR@K and achieves competitive performance on R@K. More specifically, our method outperforms the BGNN by **8.1%**, **5.9%** and **6.8%** at mR@100 on PredCls, SGCls and SGGen, respectively. These results prove that our method can effectively reduce the ambiguity of predicate representation. **2)** For other methods, our method shows excellent adaptability. It can not only significantly improve the performance of mR@K, but also improve the performance of R@K.

**Model overhead problem.** Although we utilize GPT-4 to generate object hierarchy, it is the text information generated for each category. Taking VG dataset as an example, it contains 150 object categories. The processing time for these categories is less than 5 min. Meanwhile, the text encoder of CLIP is also offline processing to extract embedding, which does not increase additional overhead.

**Do not use large models for experiments.** Taking into account the additional benefits of using large models, we also conduct experiments using WordNet and Glove. These two stable external information sources can provide stable experimental results. As shown in Tab.3, our methods still have the superior performance under this setting.

**The explanations of R@K performance.** It can be seen from Tab.1 that R@K of our method decreases slightly. It is worth mentioning that our method is based on BGNN (Li et al., 2021). Compared with this method, for PredCls task, our mR@100 increased by **8.1%**, while R@100 decreased by only **1.1%**. Thus, it does not affect the effectiveness of our method. Meanwhile, to further explore this phenomenon, we conducted a series of ablation experiments. We find that it is HCR strategy that causes the R@K decrease. Thus, for some tasks that pursue R@K, you can use only the OPHP strategy and still have SOTA performance.

| Method | Settings | | | | Module | | PredCls | | SGCls | | SGGen | |
|---|---|---|---|---|---|---|---|---|---|---|---|---|
| | $K_{gpt}$ | $K_{wordnet}$ | $W_{clip}$ | $W_{glove}$ | OPHP | HCR | mR@50/100 | R@50/100 | mR@50/100 | R@50/100 | mR@50/100 | R@50/100 |
| | - | - | - | - | - | - | 30.4/32.9 | 59.2/61.3 | 14.3/16.5 | 37.4/38.5 | 10.7/12.6 | 31.0/35.8 |
| HINet | ✓ | ✗ | ✓ | ✗ | ✓ | ✓ | **38.9/41.0** | **57.6/60.2** | **21.6/22.4** | **40.8/41.6** | **17.8/19.4** | **25.6/30.1** |
| | ✓ | ✗ | ✗ | ✓ | ✓ | ✓ | 38.4/40.5 | 57.4/59.7 | 21.2/21.9 | 40.1/41.2 | 17.6/19.2 | 25.4/29.7 |
| | ✗ | ✓ | ✓ | ✗ | ✓ | ✓ | 37.6/39.9 | 57.4/60.1 | 20.9/21.4 | 40.3/41.5 | 17.2/19.0 | 25.6/29.9 |
| | ✗ | ✓ | ✗ | ✓ | ✓ | ✓ | 37.2/39.4 | 57.1/59.5 | 20.8/21.2 | 40.1/41.1 | 17.2/18.8 | 25.2/29.6 |
| $OPHP_{only}$ | ✓ | ✗ | ✓ | ✗ | ✓ | ✗ | 35.9/37.1 | 62.0/63.1 | 19.2/19.9 | 41.2/41.9 | 15.4/16.2 | 29.2/32.1 |
| + PENET | ✓ | ✗ | ✓ | ✗ | ✓ | ✗ | 37.4/38.5 | **68.7/70.3** | 20.4/22.1 | **41.7/43.3** | 16.8/17.4 | **31.1/35.6** |
| + DRM | ✓ | ✗ | ✓ | ✗ | ✓ | ✗ | **48.1/50.8** | 44.3/46.7 | **28.2/28.6** | 30.9/32.1 | **20.9/22.8** | 22.9/25.4 |
| + RA-SGG | ✓ | ✗ | ✓ | ✗ | ✓ | ✗ | 39.2/42.9 | 62.8/65.1 | 21.8/23.2 | 38.9/40.1 | 16.7/19.1 | 26.5/31.2 |
| $HCR_{only}$ | ✓ | ✗ | ✓ | ✗ | ✗ | ✓ | 37.4/39.5 | 56.1/58.7 | 20.4/21.1 | 38.4/39.2 | 16.9/18.5 | 24.7/29.2 |
| + PENET | ✓ | ✗ | ✓ | ✗ | ✗ | ✓ | 38.4/40.6 | **64.4/67.2** | 22.6/23.1 | **37.9/39.3** | 17.2/19.1 | **25.9/30.0** |
| + DRM | ✓ | ✗ | ✓ | ✗ | ✗ | ✓ | **48.9/51.2** | 43.6/45.5 | **28.1/29.5** | 27.6/29.1 | **21.4/23.4** | 19.2/22.7 |
| + RA-SGG | ✓ | ✗ | ✓ | ✗ | ✗ | ✓ | 41.7/43.9 | 60.2/61.1 | 22.0/23.3 | 37.5/38.2 | 17.7/20.1 | 24.4/27.4 |

Table 3: The performance of our ablation study on different settings and modules on three SGG tasks. $K_{gpt}$ and $K_{wordnet}$ denote the parts obtained from GPT and WordNet, respectively. $W_{clip}$ and $W_{glove}$ denote the text embedding obtained from CLIP and Glove, respectively. The **best** and worst values are marked according to formats.

## 3.3 ABLATION STUDY

To verify the contributions of the settings and modules, we conduct the following ablation studies.

**Settings.** In general, there are two settings that need to be discussed in this paper. The first is that Eq.3 utilizes CLIP text encoder to calculate the text embedding of parts. In few shot learning, some researchers have shown that text embedding obtained by CLIP contains better semantic knowledge than Glove (Pennington et al., 2014) ($W_{clip}$). To ensure a fair comparison with existing methods, we give the experimental results obtained by utilizing CLIP/Glove to calculate text embedding ($W_{glove}$). The second is that we utilize GPT to generate object hierarchy. Considering the instability of the LLMs, we additionally use WordNet, a stable external knowledge, obtain parts (in Tab.3).

**OPHP.** We then prove the effectiveness of our OPHP strategy. This strategy aims to alleviate the ambiguity caused by utilizing a single-scale global representation to represent predicate in the previous works. We only retained this strategy for experiments. The experimental results are shown in Tab.3 ($OPHP_{only}$). Meanwhile, we also combine our OPHP strategy with other methods to verify the adaptability. We replace the method of calculating predicate representation in these papers with Eq.7 (as shown in Tab.3). These results prove that part-level features can effectively make up for the shortcomings of object-level features.

**HCR.** Finally, we prove the effectiveness of our HCR strategy. Through this strategy, we design a novel constraint for optimizing predicate representation. We only retained this strategy for experiments. The experimental results are shown in Tab.3 ($HCR_{only}$). It can be seen that this strategy can significantly improve mR@K, but also reduce R@K. The reason for this phenomenon is that our HCR strategy pays more attentions to details than the previous works. Meanwhile, we also combine our HCR strategy with other methods to verify the generalization. We add our $L_{hcr}$ (in Eq.16) to their loss function (as shown in Tab.3). These results prove that detailed decoupling of visual relationships can effectively help the model understand predicates.

## 4 CONCLUSION

In this paper, we creatively propose a method to structurally decouple the visual relationship through object hierarchy. It can effectively alleviate the ambiguity and information confounding of predicate representation in existing methods. Technically, we first construct more robust and discriminative predicate representations by dynamically fusing global object-level and local part-level representations. We then design a novel structured constraint on predicate representations by explicitly constructing correlations between object-level interactions and part-level interactions, thereby guiding the model to focus on the key information of the current interaction. Through the collaborative processing of these strategies, our HINet transcends the superficial learning of visual relations from objects and predicates, adopting a structured reasoning approach to explore their essence. Experiments have demonstrated the effectiveness of our method. Our work opens new avenues for understanding complex visual relationships and encourages future exploration.

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

## A  PROBLEM SETTING AND RELATED WORKS

### A.1  PROBLEM SETTING AND METHOD OVERVIEW

**Problem Setting.** Given an image $\mathbf{I}$, scene graph generation (SGG) aims to parse the input $\mathbf{I}$ into a scene graph $G = \{E, R\}$, where $E$ denotes the set of nodes representing objects, and $R$ represents the set of edges that encode the predicates between ordered pairs of entities. Typically, each node $e_i \in E$ is assigned a category label from a pre-defined set of object class $C_e$ and is associated with a corresponding image location indicated by a bounding box. Additionally, each edge $r_{ij} \in R$, which connects a pair of nodes $e_i$ and $e_j$, is linked to a predicate label derived from a specified set of predicate classes $C_p$ relevant to this task.

**Method Overview.** Visual relationship, commonly defined as tuple consisting of subject, predicate and object, plays an important role in visual scene understanding, Most existing works are dedicated to generating discriminative predicate representation for the detected objects, based on their

appearances, relative positions and global context. However, these global representations are inherently ambiguous and confounded, often capturing irrelevant contextual information. To address this problem, our core insight is that a seemingly holistic object-level interaction can be resolved into a set of precise part-level interactions via an object hierarchy. Compared with object-level interaction, part-level interaction not only has lower visual variability, but also provides accurate guidance for model understanding predicates. To this end, we introduce Hierarchical Inference Network (HINet). Specifically, we first construct more robust and discriminative predicate representations by dynamically fusing global object-level and local part-level representations. We then design a structured constraint on predicate representations by explicitly constructing correlations between object-level and part-level interactions, thereby guiding the model to focus on the key information of the current interaction. Through the collaborative processing of these strategies, our HINet transcends the superficial learning of visual relations from objects and predicates, adopting a structured reasoning approach to explore their essence.

## A.2 Studies of Visual Relationships

A central pursuit in computer vision is to understand the content of visual scenes. As an important part of a visual scene, the visual relationship, which describes the interaction between a subject and an object, also receives extensive attention in areas such as visual question answering (Qian et al., 2024; Lin et al., 2024; Gao et al., 2024), visual relationship detection (Li et al., 2024c; Lu et al., 2016; Liang et al., 2018), and scene graph generation (Li et al., 2024b; Zhao et al., 2024; Lin et al., 2024). One approach to visual relationship detection is to classify the entire "subject-predicate-object" triplet as a single, holistic category (Divvala et al., 2014; Ramanathan et al., 2015).

The primary challenge with this method is the combinatorial explosion of possible triplets as the number of object and predicate classes increases. This leads to a severe long-tail data distribution problem, where the model struggles to learn effective visual features for the majority of relationship categories due to a lack of sufficient training examples, while a few high-frequency relationships dominate the training process. To address this challenge, research shifts towards strategies that decouple the learning of the triplet's three components: subject, object, and predicate (Krishna et al., 2017; Tang et al., 2019; Chen et al., 2019; Zhang et al., 2019). A key work in this area by Lu (Lu et al., 2016) proposes a model that leverages language priors to aid visual relationship detection. This model first uses a pre-trained object detector to extract visual features from object pairs and then integrates a language model to predict the predicate. The language model provides statistical likelihoods of relationships (e.g., "person-riding-horse" is more common than "person-eating-horse"), effectively mitigating the data sparsity issue that arises when relying solely on visual information. This method

| Visual Genome | | |
|---|---|---|
| Objects | | |
| Category | Examples | Classes |
| Artifact | arm, tail, wheel | 32 |
| Person | boy, kid, woman | 13 |
| Clothes | cap, jean, sneaker | 16 |
| Vehicle | airplane, bike, truck | 12 |
| Flora | flower, plant, tree | 3 |
| Location | beach, room, sidewalk | 11 |
| Furniture | bed, desk, table | 9 |
| Building | building, house | 2 |
| Structure | fence, post, sign | 3 |
| Food | banana, orange, pizza | 6 |
| Part | arm, tail, wheel | 32 |
| Predicates | | |
| Geometric | above, behind, under | 12 |
| Possessive | has, part of, wearing | 8 |
| Semantic | carrying, eating, using | 24 |
| Misc | for, from, made of | 3 |

Table 4: Objects and predicates in the VG.

helps establish the dominant two-stage research paradigm: a model first detects objects and then infers pairwise predicates.

However, researchers realize that this localized perspective neglects crucial global context. For instance, a "person" and a "cake" are more likely to have an "eating" relationship in a kitchen scene but a "buying" relationship in a store. Following this insight, Scene Graph Generation (SGG) emerges as a mainstream research direction to make better use of scene context. Unlike merely detecting isolated relationship triplets, SGG aims to construct a structured graph that integrates all objects and their pairwise relationships within an image. The release of the Visual Genome dataset (Krishna et al., 2017) is a major catalyst for the field, providing large-scale, densely annotated objects, attributes, and relationships that enable the development of data-driven SGG models. Typical SGG models (Xu et al., 2017; Zellers et al., 2018) adopt a two-stage approach: first, a pre-trained object

detector (like Faster R-CNN) is used to identify objects, and then a relationship classification module is applied to all potential object pairs. These models are dedicated to generating discriminative predicate representations for the detected objects, based on their appearances, relative positions, and global context (Xu et al., 2017; Zellers et al., 2018; Li et al., 2021).

Currently, SGG research focuses on designing various message passing strategies to capture the context information of the scene. These methods typically utilize graph-based context-modeling strategies to learn discriminative representations for node and edge prediction (Li et al., 2021). A popular idea is to model the context based on a sequential model or a fully-connected graph (Xu et al., 2017; Dai et al., 2017; Li et al., 2017; Woo et al., 2018; Wang et al., 2019). In addition, some works explore sparse graph structures, which are either associated with downstream tasks or are built by trimming relationship proposals according to the category or geometry information of subject-object pairs (Tang et al., 2019; Yin et al., 2018). However, these works often rely on their specific designs for downstream tasks, which limits the flexibility of their representations. To address this problem, other works explore adaptive messaging strategies (Li et al., 2021; Kim et al., 2023; 2024). They calculate more flexible contextual representations by dynamically learning the weights of message passing, effectively capturing the contextual information of the scene.

Although these methods greatly promote the development of the field, their global representations are inherently ambiguous and confounded, often capturing irrelevant contextual information. For example, the union region of a "man" and a "bat" may simultaneously include information for "man-hold-bat," "man-wear-shirt," and "man-has-hand," making it challenging to isolate the key interaction. Meanwhile, there are only subtle differences between some similar predicates, for example, "carrying" (a person having an object in their hands) and "holding" (a person supporting an object in their hands) (Li et al., 2023). In this case, these coarse-grained global representations are difficult to provide sufficient discriminability.

## A.3 OBJECT HIERARCHY

Object hierarchy elucidates the composition of objects (Salakhutdinov et al., 2011; Deng et al., 2011). This part-object correlation provides a powerful structural prior that can be exploited to understand, recognize, and represent visual information more efficiently. For example, a "face" is composed of "eyes", a "nose", and a "mouth". By understanding these parts, a model can develop a more robust and generalizable representation of an object.

In existing works, numerous endeavors have leveraged this hierarchy to accomplish computer vision tasks (Marszałek & Schmid, 2008; Griffin & Perona, 2008; Sivic et al., 2008). In image classification tasks, some studies employ both top-down and bottom-up approaches to learn hierarchical structures (Marszałek & Schmid, 2008; Li et al., 2010), while others attempt to pre-defined such structures to help model recognize objects (Marszalek & Schmid, 2007; Verma et al., 2012; Jia et al., 2013). (Deng et al., 2012) utilizes this hierarchical framework to enhance object categories with insufficient training examples; (Liu et al., 2013) employs a pre-defined hierarchy from the Ima-

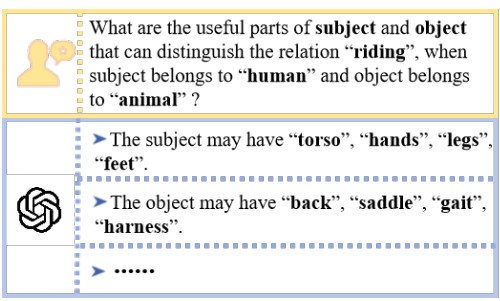

Figure 3: Examples of prompt used for generating parts.

geNet dataset to balance classification distribution. And in few-shot learning, some works (Banik et al., 2018; Tokmakov et al., 2019; Hu et al., 2019) try to enhance object representation by aggregating image features and some visual features of object components.

For visual relationships, some studies also pay attention to the help of components for the model to understand predicates (Lu et al., 2018; Tian et al., 2020; Dong et al., 2021). But these methods are completely different from our core concepts. They focus on the part itself, aiming to analyze the state of the part to enrich the semantic understanding of the image, while we aim to explore the association between objects and components, and through this association to structurally decouple the visual

relationship. From the ideological level, we rise the decoupling of objects to the decoupling of relations, which is a new perspective on SGG research.

## B  OBJECT HIERARCHY GENERATION

**The division standard of part object and non-part object.** Following the setting in Motifs (Zellers et al., 2018), we present the details of VG dataset in Tab.4. Based on it, all non-part objects are in the Tab.4 except "**Part**" (the shadow row). And in addition to what is contained in the table, component objects are complemented by WordNet or GPT-4. In this paper, we utilize GPT-4 to get the categories of parts. We realize that an object may contain a large number of parts, but only a few parts is useful for inferring visual relationships. Thus, inspired by Li et al. (2024c), we design the following prompt to get the parts (as shown in Fig.3).

## C  HIERARCHICAL CONSISTENCY REASONING

This section is a supplement of our HCR strategy, which consists of the following three components: 1) the design idea of our idea; 2) the overall structure of our idea; 3) some visualized examples.

### C.1  DESIGN IDEA

Our HCR strategy stems for the fact that although all visual relationships are compose of objects and predicates, some of them can be effectively represented by the combinations of others. In this paper, we summarize this phenomenon as a visual relationship hierarchy caused by object hierarchy.

On the one hand, the interaction between parts is usually the key to our recognition of visual relationships. For example, we judge "man-stand on-rood" by observing "foot-on-surface". But for a pair of non-part objects, the interaction between parts is obviously not enough. We can never infer "man-stand on-road" from "foot-on-surface" alone (it could be some other answers, such as: "woman-stand on-floor"). Therefore, we propose a question: *how to infer object-predicate-object from pat-predicate-part*? The answer is obvious: we need to determine which object these parts belong to. So far, we have obtained a reasoning way for us to understand visual relationships. For object-predicate-object, part-predicate-part represents the key for us to identify it and object-predicate-part helps us to establish the relationship between parts and objects.

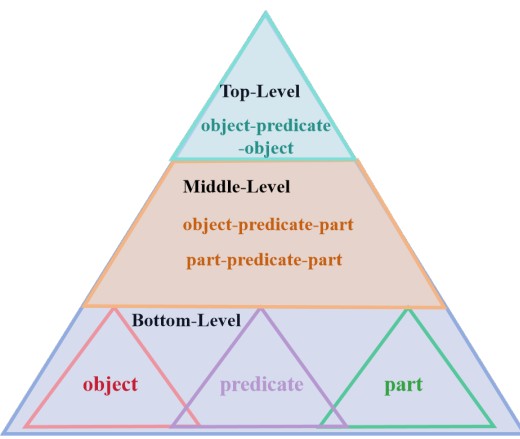

Figure 4: The example of our visual relationship hierarchy.

On the other hand, the associations between objects and parts indicate that there are possessive relations between objects and parts. It means that we can always find a set of object-predicate-part, part-predicate-part to represent object-predicate-object. Meanwhile in the previous works, visual relationship is defined as: visual relationship is a pair of localized objects via a predicate (Lu et al., 2016). From the hierarchical knowledge perspective (Deng et al., 2011; Salakhutdinov et al., 2011), objects and predicates are treated as primitives, which are located the lowest level; object-predicate-object is located at a highest level. This structure is essentially a detailed decoupling method for visual relationship, and can help the model to infer visual relationship more accurately.

### C.2  OVERALL STRUCTURE

Inspired by the procedure described above, we extend object hierarchy to a visual relationship hierarchy to reveal the correlations between different visual relationships. Specifically, it contains three levels: bottom-level with objects and predicates, top-level with object-predicate-object, and

| Images | Reasoning Rules | |
|---|---|---|
| | bottom to others | middle-to-top |
| | woman-hold-umbrella = woman + hold + umbrella | woman-hold-umbrella = woman-has-hand + hand-hold-handle + handle-of-umbrella |
| | man-touch-horse = man + touch + horse | man-touch-horse = man-has-hand + hand-touch-neck + neck-of-horse |
| | bus-on-road = bus + on + road | bus-on-road = bus-has-wheel + wheel-on-surface + surface-of-road |

Figure 5: Some examples of the detailed reasoning rules in our visual relationship hierarchy.

middle-level with object-predicate-part and part-predicate-part (as shown in Fig.4). In this structure, higher-level entities can be represented by a combination of lower-level entities.

For humans, it seems that we do not need to understand visual relationships in this way. It is because this way is simple and intuitive for us. But for a machine, it doesn't have the ability to find this way on its own, nor does it have enough data to learn every visual relationship. It means that we need to guide the model to understand visual relationships. We show some visualized results of our reasoning rules in Fig.5.

## D    EXPERIMENTS

In general, to fully verify the effectiveness of our idea, we conducted experiments on four datasets: Visual Genome (Krishna et al., 2017), Open Images V6 (Kuznetsova et al., 2020), V-COCO (Gupta & Malik, 2015) and HICO-DET (Chao et al., 2018). In our paper, we have reported the results of Visual Genome. Thus in this section, we show the experimental results of Open Images V6, V-COCO and HICO-DET. It is worth mentioning that HOI has been very different from SGG after the long-term development. Thus, we can only improve TIN (Li et al., 2019) to verify the effectiveness of our idea. These experimental results do not have the ability to compare with the advanced models in HOI. We show some visualized result in Fig.6.

| Method | mR@50 | R@50 | $wmAP_{rel}$ | $wmAP_{phr}$ | $score_{wtd}$ |
|---|---|---|---|---|---|
| VCTree | 33.9 | 74.1 | 34.2 | 33.1 | 40.2 |
| RelDN | 37.2 | 75.3 | 32.2 | 33.4 | 42.0 |
| Motifs | 32.7 | 71.6 | 29.9 | 31.6 | 38.9 |
| BGNN | 40.5 | 75.0 | 33.5 | 34.1 | 42.1 |
| HetSGG | 42.7 | 76.8 | 34.6 | 35.5 | 43.3 |
| Unbiased | 35.5 | 69.3 | 30.7 | 32.8 | 39.3 |
| PENET | - | 76.5 | 36.6 | 37.4 | 44.9 |
| EdgeSGG | 43.3 | **77.1** | 36.4 | 37.4 | 44.9 |
| HINet | **45.1** | 76.9 | **37.7** | **38.4** | **45.8** |

Table 5: Performance comparison with the SoTA methods on Open Images V6 dataset. The **best** method is marked according to formats.

### D.1    EXPERIMENTS OF OPEN IMAGES V6

**Dataset Details.** Open Images V6 dataset is a large-scale dataset commonly used for SGG tasks (Kuznetsova et al., 2020). It contains a diverse collection of over 133k images with 126,368 training, 1,813 validation, and 5,322 testing images. This dataset provides object-level annotations for each

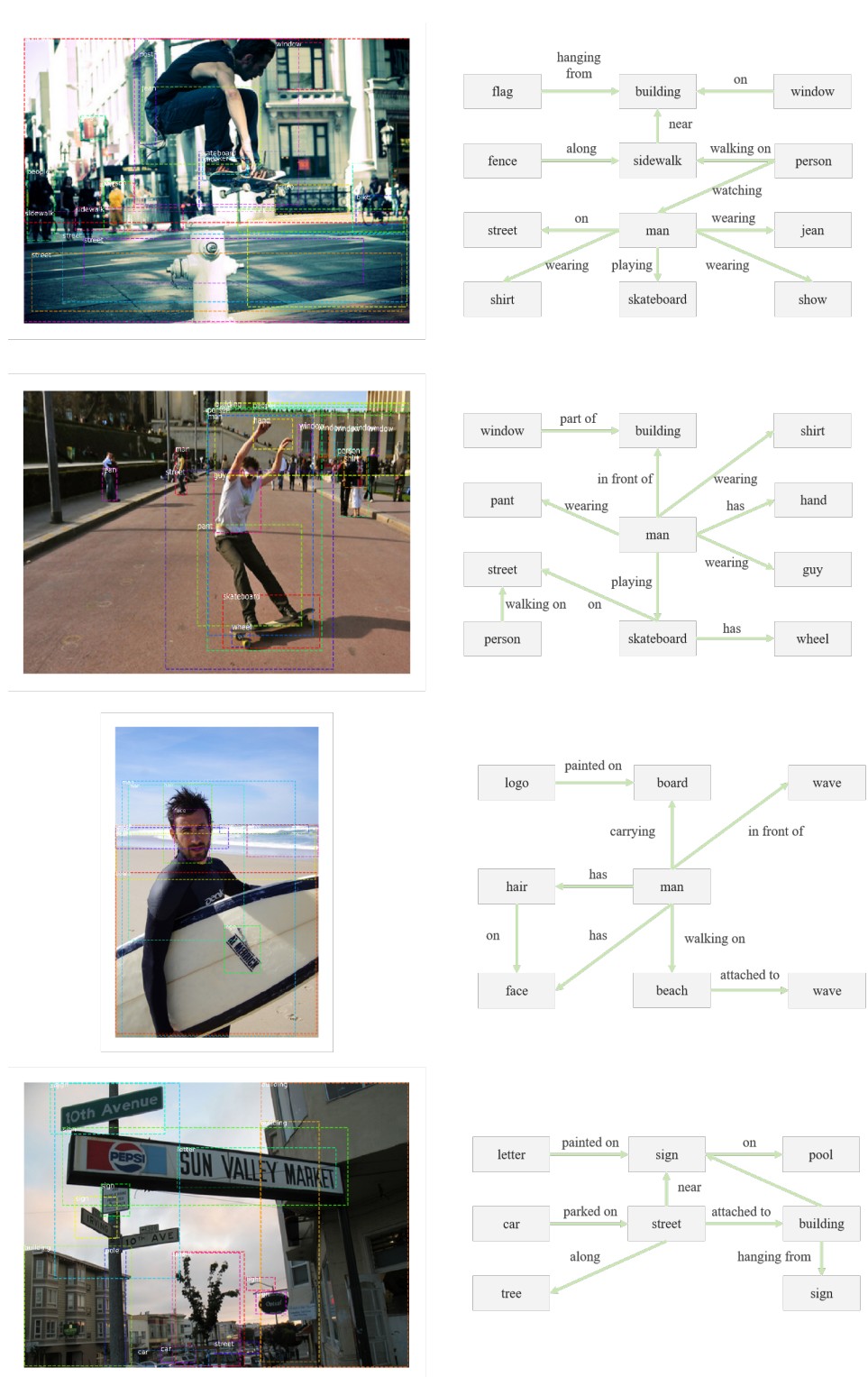

Figure 6: The visualized results of our method.

image, including bounding boxes and 301 object categories. In addition, it includes 31 relationship annotations that describe the interactions between pairs of objects within a scene.

**Metrics.** For this dataset, we follow the same data processing and evaluation protocols in the existing works (Li et al., 2021; Zhang et al., 2019). The mR@50, R@50, weighted mean AP of relationships ($wmAP_{rel}$), and weighted mean AP of phrase ($wmAP_{phr}$) are used as evaluation metrics. Following standard evaluation metrics of Open Images refers to (Li et al., 2021; Zhang et al., 2019), the weight metric $score_{wtd}$ is computed as: $score_{wtd} = 0.2 \times R@50 + 0.4 \times wmAP_{rel} + 0.4 \times wmAP_{phr}$.

**Quantitative Results.** The quantitative results are shown in Tab.5. Our method achieves the SOTA performance on mean recall and competitive results on weighted metric score. These results demonstrate the effectiveness of our method.

| Method | Default | | | Known Object | | |
|---|---|---|---|---|---|---|
| | Full | Rare | Non-Rare | Full | Rare | Non-Rare |
| HO-RCNN (Chao et al., 2018) | 7.81 | 5.37 | 8.54 | 10.41 | 8.94 | 10.85 |
| InteractNet (Gkioxari et al., 2018) | 9.94 | 7.16 | 10.77 | - | - | - |
| GPNN (Qi et al., 2018) | 13.11 | 9.34 | 14.23 | - | - | - |
| iCAN (Gao et al., 2018) | 14.84 | 10.45 | 16.15 | 16.26 | 11.33 | 17.73 |
| TIK (Li et al., 2019) | 17.22 | 13.51 | 18.32 | 19.38 | 15.38 | 20.57 |
| HINet | 20.46 | 15.33 | 21.51 | 22.71 | 16.15 | 23.23 |

Table 6: Results comparison on HICO-DET.

## D.2 HOI

**HICO-DET.** HICO-DET (Chao et al., 2018) includes 47,776 images, including 38,118 in train set, 9658 in test set, 600 HOI categories on 80 object categories and 117 verbs, and provides more than 150k annotated human-object pairs.

**V-COCO.** V-COCO (Gupta & Malik, 2015) includes 10,346 images, including 2,553 in train set, 2,867 in validate set, 4,946 in test set, and 16,199 person instances. Each person has annotations for 29 action categories. The objects are divided into two types: "objects" and "instrument".

**Metrics.** Following the settings adopted in (Chao et al., 2018; Li et al., 2019), the role mean average precision is used to measure the performance.

## D.3 RESULTS

| Method | $AP_{role}$ |
|---|---|
| InteractNet | 40.0 |
| GPNN | 44.0 |
| iCAN | 44.7 |
| TIN | 47.8 |
| HINet | 49.6 |

Table 7: Results comparison on V-COCO.

The result of our model in HICO-DET is shown in Tab.6 and the result of our model in V-COCO is shown in Tab.7. For SGG task, VG dataset and Open Images V6 dataset are more representative. The improvement of M@K and mR@K is enough to show that our method is effective for most predicates. However, these datasets contain very few similar predicates. Thus, we conduct experiments on two datasets of HOI. Extensive experimental results demonstrate the effectiveness of our method. Our work opens new avenues for understanding complex visual relationships and encourages future exploration.

## E LLMS USAGE DESCRIPTION

Our experiment uses LLM to generate object hierarchy, and the specific configuration can be found in Sec.3.1. In the process of writing, GPT-4o and Gemini are used for polishing, but only as an auxiliary tool.

