# OpenReview forum: "Object Hierarchy Guides Visual Relationship Reasoning"
_ICLR.cc/2026/Conference — ICLR 2026 Conference Withdrawn Submission_

### Official Review · Reviewer_R4mz · 2025-10-23

**Soundness:** 3
**Presentation:** 4
**Contribution:** 4
**Rating:** 6
**Confidence:** 2

**Summary:**

This paper proposes HINet (Hierarchical Inference Network) for Scene Graph Generation (SGG), which leverages object hierarchy to improve visual relationship understanding. It decomposes object-level interactions into part-level interactions, thereby providing more discriminative and less ambiguous predicate representations. The method introduces two main strategies: Object-Part Hybrid Perception (OPHP) for fusing multi-scale features and Hierarchical Consistency Reasoning (HCR) for structured constraint learning. The model is validated on several benchmarks, including Visual Genome, Open Images V6, V-COCO, and HICO-DET, showing significant improvements over existing SOTA methods on metrics like mR@K (mean recall).

**Strengths:**

1. The idea of decomposing object-level relationships into part-level interactions via an object hierarchy is interesting and well-motivated. The concept of a visual relationship hierarchy offers a fresh perspective for understanding predicates.
2. The two strategies (OPHP and HCR) that translate the core idea into the HINet model are clear and well-designed. The OPHP strategy's use of part-level text embeddings to guide attention for extracting relevant visual features is a clever implementation. The HCR strategy creatively introduces the concept of translation invariance from knowledge graphs into visual relationship modeling, imposing a strong structural prior that compels the model to learn the compositional rules of underlying relationships.
3. The paper is evaluated on multiple datasets (Visual Genome, Open Images V6, V-COCO, HICO-DET), showing consistent improvements, especially on the mean recall (mR@K) metric. It also demonstrates good generalization by successfully integrating with several existing SGG models (PENET, DRM, RepSGG, RA-SGG).

**Weaknesses:**

1. The method heavily relies on GPT-4 for generating part categories. While the WordNet alternative is presented, a more systematic evaluation is somewhat lacking. The paper would be more persuasive if it further demonstrated the impact of hierarchy quality on performance (e.g., an analysis of hierarchy accuracy vs. mR@K).
2. The paper shows consistent improvements in mR@K but a decrease in R@K (Tables 1 and 2). Although the authors suggest using only the OPHP strategy for tasks that prioritize R@K, the underlying reason for this trade-off warrants a deeper analysis. Why does imposing a structured constraint on rare relationships hurt the model's performance on common ones?
3. Compared to baseline models like BGNN, HINet appears to introduce significant computational complexity. For each relationship proposal, the model needs to generate attention maps for all parts of an object, extract fine-grained features, and compute additional representations for the hierarchical consistency loss. Providing an analysis of the extra computational costs (e.g., training/inference time, FLOPs) would make the paper more complete.

**Questions:**

See Weaknesses

---

### Official Review · Reviewer_nkbM · 2025-10-31

**Soundness:** 3
**Presentation:** 3
**Contribution:** 3
**Rating:** 4
**Confidence:** 4

**Summary:**

This paper presents an effective method for visual relationship reasoning by structurally decoupling object-level interactions into part-level ones via object hierarchy. The proposed HINet, with its two core strategies (OPHP and HCR), demonstrates strong performance, particularly on tail predicates, and shows good generalizability across multiple models and benchmarks.

**Strengths:**

1. The idea of using object-part hierarchies for predicate reasoning is intuitively appealing. It draws inspiration from human visual reasoning.

2. The descriptions of OPHP and HCR are mathematically detailed, with clear equations and figures.

3. The proposed architecture is model-agnostic and can be integrated into existing SGG models.

**Weaknesses:**

1. The method relies on GPT-4 and CLIP for hierarchy generation and embeddings. Although the authors claim it adds minimal overhead, it raises fairness concerns for comparisons.

2. While the integration is novel, the idea of hierarchical decomposition and structured consistency resembles prior works (e.g., hierarchical SGG models such as HIKER-SGG and HIERCOM). The paper could better clarify its distinction (lines 95-98).

3. The multi-level reasoning framework adds several intermediate representations and loss terms. It may complicate training and interpretation, and a detailed time complexity/training time comparison is lacking.

**Questions:**

1. Can you show some intuitive examples of the ambiguous global object-level representations? Similarly, can you show examples where part-level reasoning corrected errors made by object-level methods?

2. How does the model decide when to rely on global vs. part-level reasoning?

3. The principles of "Bottom-to-others" and "Middle-to-top" are artificially designed. Can they cover all cases?

4. How deep is the hierarchy (object → part → sub-part)? Is performance sensitive to the level of granularity?

---

### Official Review · Reviewer_LkAp · 2025-10-31

**Soundness:** 3
**Presentation:** 3
**Contribution:** 3
**Rating:** 4
**Confidence:** 5

**Summary:**

This paper addresses the ambiguity and information confounding issues of predicate representation in visual relationship reasoning by leveraging object hierarchy. It proposes a Hierarchical Inference Network (HINet) with two core strategies: Object-Part Hybrid Perception (OPHP) for fusing global and local features, and Hierarchical Consistency Reasoning (HCR) for structured constraints. The model is validated on four datasets (Visual Genome, Open Images V6, V-COCO, HICO-DET) and achieves competitive or state-of-the-art performance on scene graph generation (SGG) tasks. Additionally, it exhibits model-agnostic properties, enabling integration with existing SGG models to enhance their performance. The key contributions lie in the novel structural decoupling idea via object hierarchy, the design of HINet, and extensive experimental validations.

**Strengths:**

(1) The core insight of resolving object-level interactions into part-level interactions via object hierarchy is innovative, effectively addressing the long-standing ambiguity and confounding problems in predicate representation.

(2) The dual-strategy design (OPHP and HCR) is well-coordinated: OPHP enriches predicate features by fusing multi-level representations, while HCR imposes structured constraints, forming a comprehensive solution for visual relationship reasoning.

(3) Extensive experiments on multiple datasets (covering SGG and HOI tasks) and sufficient ablation studies demonstrate the effectiveness, generality, and robustness of the proposed method.

(4) The model-agnostic nature allows easy integration with existing SGG frameworks, enhancing practical value and promoting wider adoption in the field.

**Weaknesses:**

(1) The implementation details of object hierarchy generation are insufficient. While the paper mentions using GPT-4 and WordNet to collect part categories, the specific prompt engineering for GPT-4 (only referenced to Sec.B without detailed content) and the criteria for filtering "useful parts" are unclear, hindering reproducibility.

(2) The calculation of relevance scores in the Representation Aggregation (RA) step of OPHP is not fully specified. The paper states that relevance scores measure the importance of parts for interactions, but the mathematical formulation, loss function, and training process for these scores are missing, leading to ambiguity about how the model learns part relevance.

(3) The theoretical foundation of the visual relationship hierarchy in HCR is weak. The paper claims higher-level relationships can be represented by lower-level ones, but lacks formal proof or empirical justification for the universality of this assumption (e.g., whether it holds for complex predicates like "interact with" or "depend on").

(4) The explanation for the slight decrease in R@K performance is inadequate. The paper attributes it to the HCR strategy's focus on details, but fails to analyze which types of predicates (common vs. tail) are most affected, and whether there is a trade-off adjustment mechanism to balance R@K and mR@K.

(5) The comparison with state-of-the-art (SOTA) methods is incomplete. For HOI tasks, the paper only improves TIN (Li et al., 2019) and admits it cannot compete with advanced HOI models, but does not explain why it avoids comparing with recent SOTA HOI methods (e.g., those from CVPR 2023-2024), limiting the assessment of cross-task generality.

(6) The choice of hyperparameters lacks justification. For example, the weights λ₀ and λₕcr in the total loss function are set to 1 and 5 respectively, but the paper provides no analysis of how different weight combinations affect model performance, nor does it conduct a sensitivity analysis.

(7) The handling of part-level objects is ambiguous. The paper classifies objects into part-level and non-part-level, but does not clarify how part-level objects (e.g., "hand") are processed in OPHP and HCR--whether their own part hierarchies are considered or ignored.

(8) The text embedding methods (GloVe and CLIP) are not thoroughly compared. The paper mentions CLIP provides better semantic knowledge but does not quantify the performance gap between the two embeddings across different tasks/datasets, nor explains why CLIP is chosen as the default.

(9) The visualization results are insufficient. Figure 6 only shows qualitative examples without quantitative analysis of how well the model captures part-level interactions, making it hard to verify the effectiveness of structural decoupling.

(10) The computational overhead analysis is superficial. While the paper states that GPT-4 processing takes less than 5 minutes for 150 categories, it does not report the additional inference time introduced by OPHP and HCR compared to baseline models (e.g., BGNN), which is critical for practical applications.

(11) The generalization to zero-shot or few-shot scenarios is unexplored. Given the paper's focus on predicate representation and part-level features, it does not test whether HINet can improve performance in low-data regimes, a key challenge in SGG.

(12) The paper does not discuss potential failures of the proposed method. For example, are there cases where object hierarchy is unavailable or ambiguous (e.g., abstract objects without clear parts), and how does HINet handle such scenarios?

(13) The related work section lacks a comprehensive comparison with hierarchical SGG methods. The paper distinguishes itself from methods that model predicate hierarchy or use object parts implicitly, but does not compare with recent works like Hiker-SGG (Zhang et al., 2024) that also use hierarchical knowledge, leading to an incomplete positioning of the proposed method.

(14) The part proposal generation in HCR's Middle-to-top strategy is unclear. The paper mentions using RegionCLIP but does not specify the training details, inference accuracy, or impact of part proposal quality on HCR's performance.

(15) The loss function design for HCR is not fully justified. The paper introduces Lᵦₜₒ and Lₘₜₜ, but does not explain why cosine similarity maximization (for Lᵦₜₒ) and additive feature alignment (for Lₘₜₜ) are chosen over other alternatives (e.g., Euclidean distance minimization).

(16) The paper does not address data bias in object hierarchy generation. GPT-4 may generate biased part categories (e.g., over-representing common parts), but the paper does not evaluate how this bias affects model performance across different object categories.

(17) The interaction between OPHP and HCR is not analyzed. The ablation study shows both strategies contribute to performance, but does not explore whether they have complementary or redundant effects, nor whether the order of applying them matters.

(18) The evaluation metrics lack diversity. For SGG tasks, the paper only uses recall-based metrics (R@K, mR@K, M@K) and does not report precision-based metrics (e.g., Precision@K) or F1-scores, which are important for assessing model reliability in real-world applications.

(19) The paper does not compare with end-to-end SGG models. Most comparisons are with two-stage models, but recent end-to-end methods (e.g., DSG G) have shown promising results--omitting these comparisons limits the assessment of HINet's competitiveness in modern SGG paradigms.

(20) The part selection process in HCR's Middle-to-top strategy is arbitrary. For predicates not strongly related to specific parts, the paper uses random part selection, but does not evaluate how this randomness affects model stability or performance.

(21) The paper lacks analysis of model complexity. It does not report the number of parameters, FLOPs, or memory usage of HINet compared to baselines, making it difficult to assess the trade-off between performance and computational cost.

(22) The semantic feature calculation for objects is not detailed. The paper mentions using class labels to compute semantic features (wᵢ) but does not specify the method (e.g., one-hot encoding, pre-trained word embeddings), leading to reproducibility issues.

(23) The paper does not discuss the impact of object detection accuracy on HINet. Since HINet relies on object proposals from Faster R-CNN, it is unclear how errors in object detection (e.g., missed detections, false positives) propagate to predicate reasoning performance.

(24) The generalization to cross-dataset scenarios is untested. The paper validates on four datasets but does not conduct cross-dataset evaluation (e.g., training on Visual Genome and testing on Open Images V6), which is crucial for demonstrating the model's robustness to domain shifts.

(25) The paper does not provide qualitative analysis of similar predicates. It claims HINet can distinguish "carrying" and "holding," but lacks side-by-side examples of how the model's part-level features capture the subtle differences between these predicates.

(26) The LLM usage description is insufficient. The paper mentions using GPT-4o and Gemini for writing polishing but does not clarify whether these tools were used in designing experiments or analyzing results, raising concerns about potential ethical issues or methodological biases.

(27) The paper does not address the scalability of object hierarchy generation. For datasets with thousands of object categories (e.g., ImageNet), it is unclear whether GPT-4 can efficiently generate part categories, or if there is a scalable alternative.

(28) The paper lacks a discussion on the interpretability of HINet. While the model uses part-level features and structured constraints, it does not show how to interpret the model's decisions (e.g., which parts contribute most to a specific predicate prediction).

(29) The paper does not compare with language-augmented SGG models. Recent works (e.g., zero-shot visual relation detection using LLMs) integrate language knowledge, but HINet's use of text embeddings is not compared with these methods, limiting the assessment of its language-vision integration effectiveness.

(30) The paper's experimental setup for Visual Genome lacks details. It mentions using 150 frequent object categories and 50 frequent predicates, but does not specify how these categories are selected (e.g., frequency thresholds), leading to potential inconsistencies in reproducibility.

(31) The paper does not report the standard deviation of experimental results. All performance metrics are presented as single values without error bars, making it difficult to assess the stability of HINet's performance across multiple runs.

(32) The paper's handling of background categories is unclear. In predicate classification, it mentions foreground and background categories, but does not specify how background categories are defined or how they affect the loss function and evaluation.

(33) The paper does not explore alternative fusion strategies in OPHP. The RA strategy uses weighted averaging of fine-grained features, but does not compare with other fusion methods (e.g., attention mechanisms, concatenation without averaging) to justify its choice.

(34) The paper's HCR strategy is compared to TransE but lacks a detailed analogy. It does not explain how the visual relationship hierarchy maps to knowledge graph structures, nor does it leverage insights from knowledge graph embedding to improve HCR.

(35) The paper does not test the impact of part number on performance. For objects with more parts (e.g., "car" vs. "apple"), it is unclear whether increasing part count improves or degrades predicate representation quality.

(36) The paper's validation set usage is unclear. It mentions following standard data splits but does not specify whether the model is tuned on the validation set (e.g., hyperparameter selection), which could affect the fairness of comparisons with other methods.

(37) The paper does not address the long-tail problem in predicate distribution. While mR@K is improved, it does not analyze whether HINet specifically benefits tail predicates more than common ones, or if the improvement is uniform across predicate frequencies.

(38) The paper's geometric feature calculation is not detailed. It mentions using bounding boxes to compute geometric features (gᵢ) but does not specify the exact formulation (e.g., relative position, size ratio), leading to reproducibility issues.

(39) The paper does not compare with multi-scale feature fusion methods. OPHP's fusion of global and local features is similar to existing multi-scale approaches, but the paper does not distinguish itself from these methods, limiting the novelty of OPHP.

(40) The paper's conclusion overstates its contributions. It claims to "open new avenues for understanding complex visual relationships" but does not discuss future directions or unresolved issues, making the contribution claim less convincing.

**Questions:**

**To facilitate discussions during the Rebuttal phase, authors are advised to respond point-by-point (indicating the question number).**

(1) Could you provide the full prompt used for GPT-4 to generate part categories (referenced in Sec.B) and explain the criteria for filtering "useful parts" for visual relationship reasoning?

(2) Please detail the mathematical formulation and training process of the relevance scores in the RA step of OPHP. How are these scores optimized to reflect part importance for specific interactions?

(3) Can you provide formal proof or additional empirical evidence (e.g., statistics on how many visual relationships follow the hierarchy) to justify the universality of the visual relationship hierarchy assumption in HCR?

(4) For the slight decrease in R@K, could you analyze the performance of HINet on common vs. tail predicates separately? Is there a way to adjust HCR (e.g., hyperparameter tuning) to balance R@K and mR@K?

(5) Why did you only improve TIN for HOI tasks instead of comparing with recent SOTA HOI methods (e.g., from 2023-2024)? Please supplement cross-task comparisons to demonstrate HINet's generality.

(6) Could you conduct a sensitivity analysis of the hyperparameters λ₀ and λₕcr, and explain why the chosen values (1 and 5) are optimal?

(7) How are part-level objects (e.g., "hand") processed in OPHP and HCR? Do you consider their internal part hierarchies, or treat them as atomic entities?

(8) Please provide a detailed comparison of GloVe and CLIP text embeddings across all datasets/tasks. What specific advantages does CLIP offer that justify its use as the default?

(9) Could you supplement quantitative analysis of the visualization results (Fig.6), such as the accuracy of part-level interaction detection, to verify the effectiveness of structural decoupling?

(10) Please report the additional inference time, parameter count, and FLOPs of HINet compared to baseline models (e.g., BGNN) to assess computational overhead.

(11) Have you tested HINet's performance in zero-shot or few-shot SGG scenarios? If not, could you supplement these experiments to demonstrate generalization to low-data regimes?

(12) Can you provide examples of scenarios where object hierarchy is unavailable/ambiguous, and explain how HINet handles them? What is the model's performance in such cases?

(13) Please compare HINet with recent hierarchical SGG methods like Hiker-SGG (Zhang et al., 2024) in terms of methodology, performance, and computational cost.

(14) Could you detail the training process and accuracy of part proposal generation using RegionCLIP in HCR's Middle-to-top strategy? How does part proposal quality affect HCR's performance?

(15) Why did you choose cosine similarity maximization for Lᵦₜₒ and additive feature alignment for Lₘₜₜ? Have you tested alternative loss formulations (e.g., Euclidean distance) and what were the results?

(16) How does the bias in GPT-4-generated part categories affect model performance across different object categories? Could you provide a quantitative analysis of this bias?

(17) Do OPHP and HCR have complementary or redundant effects? Could you conduct experiments to test the performance of OPHP and HCR in different orders (e.g., OPHP first vs. HCR first)?

(18) Please supplement precision-based metrics (e.g., Precision@K) and F1-scores for all datasets/tasks to provide a more comprehensive evaluation of HINet's performance.

(19) Could you compare HINet with recent end-to-end SGG models (e.g., DSG G) to demonstrate its competitiveness in modern SGG paradigms?

(20) How does random part selection in HCR's Middle-to-top strategy affect model stability? Could you report the standard deviation of performance across multiple random seeds?

(21) Please provide detailed model complexity analysis (parameters, FLOPs, memory usage) compared to baselines to assess the performance-computational cost trade-off.

(22) Could you specify the method used to compute semantic features (wᵢ) for objects (e.g., pre-trained embeddings, one-hot encoding)?

(23) How do object detection errors (missed detections, false positives) affect HINet's predicate reasoning performance? Could you provide a sensitivity analysis?

(24) Have you conducted cross-dataset evaluation (e.g., train on Visual Genome, test on Open Images V6)? Please supplement these results to demonstrate domain adaptation capability.

(25) Could you provide qualitative examples of how HINet distinguishes similar predicates (e.g., "carrying" vs. "holding") using part-level features?

(26) Please clarify the extent of GPT-4o and Gemini's usage in the paper (e.g., only writing polishing vs. experiment design) to address potential ethical concerns.

(27) How scalable is HINet's object hierarchy generation for datasets with thousands of object categories? Is there a lightweight alternative to GPT-4?

(28) Could you add interpretability analyses (e.g., attention maps showing key parts for predicate predictions) to explain HINet's decision-making process?

(29) Please compare HINet with language-augmented SGG models (e.g., LLM-based zero-shot methods) to demonstrate the effectiveness of its text embedding integration.

(30) Could you specify the frequency thresholds used to select 150 object categories and 50 predicates for Visual Genome experiments?

(31) Please report the standard deviation of performance metrics across multiple experimental runs to demonstrate HINet's stability.

(32) How are background categories defined in predicate classification, and how do they affect the loss function and evaluation?

(33) Could you compare the RA fusion strategy with alternative methods (e.g., attention mechanisms) to justify its effectiveness?

(34) How does the visual relationship hierarchy in HCR map to knowledge graph structures? Could you leverage knowledge graph embedding insights to improve HCR?

(35) How does the number of parts per object affect HINet's performance? Could you test objects with varying part counts (e.g., simple vs. complex objects)?

(36) Did you tune hyperparameters on the validation set? If so, please detail the tuning process to ensure fair comparisons with other methods.

(37) Could you analyze HINet's performance on tail vs. common predicates separately to demonstrate its effectiveness in addressing the long-tail problem?

(38) Please specify the exact formulation of geometric features (gᵢ) computed from bounding boxes (e.g., relative position, size ratio).

(39) How does OPHP differ from existing multi-scale feature fusion methods in SGG? Please provide a detailed comparison to highlight its novelty.

(40) Could you discuss future directions or unresolved issues of HINet to support the claim of "opening new avenues" in the conclusion?

---

### Official Review · Reviewer_k7c5 · 2025-11-01

**Soundness:** 2
**Presentation:** 1
**Contribution:** 2
**Rating:** 0
**Confidence:** 5

**Summary:**

This paper proposes a Hierarchical Inference Network (HINet) for visual relationship detection. The core idea is to address the ambiguity of global object-level representations by leveraging an object hierarchy to incorporate more precise part-level interactions. The method involves dynamically fusing object-level and part-level features and applying a structured constraint to guide the model. The authors claim the approach is effective and versatile, enhancing existing models.

**Strengths:**

1. Clarity in Technical Description:A notable strength of this paper is the clear and relatively detailed description of the proposed HINet architecture. The components, including the dynamic fusion mechanism and the structured constraint, are explained in a way that makes the methodology understandable and potentially reproducible. This level of technical clarity is commendable.

**Weaknesses:**

1. Perceived Lack of Novelty and Outdated Approach:The central technique—using object hierarchies and part-level information as a form of prior knowledgeto reason about fine-grained relationships—is a well-established direction in SGG and computer vision. Methods exploring part-whole hierarchies, visual phrases, and compositional reasoning have been explored extensively in the past. The paper does not adequately position itself against this existing body of work or convincingly argue why this particular instantiation represents a significant conceptual advance over prior art. The overall framework feels like a refinement of older ideas rather than a novel paradigm.

2. Insufficient and Outdated Experimental Comparisons:This is a critical flaw. The field of SGG evolves rapidly. To demonstrate the competitiveness of a new method, it is essential to compare it against the most recent state-of-the-art approaches (e.g., from 2024, and ideally 2025 if available). The absence of such comparisons raises serious doubts about the method's current relevance and performance. The reader is left wondering if the reported improvements are marginal compared to modern baselines.

3. Poorly Presented Figures and Motivation:The figures fail to effectively illustrate the method's motivation or architectural details.

**Questions:**

1. Novelty and Related Work:The idea of using part-level hierarchies is not new. Could you more explicitly define the key technical noveltyof HINet compared to previous part-based or hierarchical models for SGG? Specifically, what is fundamentally new about your "dynamic fusion" and "structured constraint" beyond existing feature fusion and regularization techniques? Please position HINet more precisely within the landscape of existing literature.

2. Experimental Benchmarking:It is essential to include comparisons with the most recent state-of-the-art methods (from 2024-2025). Please update your results tables to include these strong, contemporary baselines. Without this, it is impossible to assess the true contribution of your work.

---

### Note · Authors · 2025-11-14

I have read and agree with the venue's withdrawal policy on behalf of myself and my co-authors.